# Differentiable Model Compression via Pseudo Quantization Noise

**Alexandre Défossez**[*]                                    *defossez@fb.com*
*Meta AI, FAIR Team, Paris, France*

**Yossi Adi**[*]                                             *adiyoss@fb.com*
*Meta AI, FAIR Team, Tel-Aviv, Israel*

**Gabriel Synnaeve**                                        *gab@fb.com*
*Meta AI, FAIR Team, Paris, France*

**Reviewed on OpenReview:** *https://openreview.net/forum?id=DijnKziche*

## Abstract

We propose DIFFQ a differentiable method for model compression for quantizing model parameters without gradient approximations (e.g., *Straight Through Estimator*). We suggest adding independent pseudo quantization noise to model parameters during training to approximate the effect of a quantization operator. DIFFQ is differentiable both with respect to the unquantized weights and the number of bits used. Given a single hyper-parameter balancing between the quantized model size and accuracy, DIFFQ optimizes the number of bits used per individual weight or groups of weights, in end-to-end training. We experimentally verify that our method is competitive with STE based quantization techniques on several benchmarks and architectures for image classification, language modeling, and audio source separation. For instance, on the ImageNet dataset, DIFFQ compresses a 12 layers transformer-based model by more than a factor of 8, (lower than 4 bits precision per weight on average), with a loss of 0.3% in model accuracy. Code is available at github.com/facebookresearch/diffq.

## 1 Introduction

An important factor in the adoption of a deep learning model for real-world applications is how easily it can be pushed to remote devices. It has been observed that larger models usually lead to better performance, for instance with larger ResNets (He et al., 2016) achieving higher accuracies than smaller ones. In response, the community has worked toward smaller, and more efficient models (Tan & Le, 2019). Yet an EfficientNet-B3 is still almost 50MB, a considerable amount if the model is to be included in online applications, or updated with limited network capabilities. For other applications, such as language modeling (Vaswani et al., 2017) or source separation (Défossez et al., 2019), the typical model size is closer to 1GB, ruling out any kind of mobile usage. Efficient model compression is thus important for on device adoption of deep learning models. Thus, we focus in the present work on reducing model size, rather than achieving computational gains.

The simplest method to reduce model size consists in decreasing the number of bits used to encode individual weights. For instance, using 16 bits floating point numbers halves the model size, while retaining a sufficient approximation of the set of real numbers, $\mathbb{R}$, to train with first-order optimization methods (Micikevicius et al., 2018). When considering lower precision, for instance, 8 or 4 bits, the set of possible values is no longer a good approximation of $\mathbb{R}$, hence preventing the use of first-order optimization methods. Specifically, uniform quantization requires using the `round` function, which has zero gradients wherever it is differentiable.

Quantization can be done as a post-processing step to regular training. However, errors accumulate in a multiplicative fashion across layers, with a possibly uncontrolled decrease in the model accuracy. Courbariaux et al.

---

[*]Equal contribution.

(2016) and later Krishnamoorthi (2018) propose to use a gradient Straight-Through-Estimator (STE) (Bengio et al., 2013) in order to provide a non-zero gradient to the original weights. This allows the model to adapt to quantization during training and reduces the final degradation of performance. However, Fan et al. (2021) noticed instability and bias in the learned weights, as STE is not the true gradient to the function.

The nature of quantization noise has been extensively studied as part of Analog-to-Digital Converters (ADC). In particular, a useful assumption to facilitate the design of post-processing filters for ADC is the independence of the input value and the "Pseudo Quantization Noise" (PQN), as formalized by Widrow et al. (1996). In this work, we show that it also applies to deep learning model quantization, and provides a simple framework in which the output and the quantized model size are both differentiable, without any use of STE. This allows to optimally set the number of bits used per individual weight (or group of weights) to achieve a trade-off between size and accuracy, in a single training and at almost no extra cost. Even when the number of bits to use is fixed, we show that unlike STE, using independent pseudo quantization noise does not introduce bias in the gradient and achieves higher performance. Although PQN has been proposed before for quantization (Baskin et al., 2018a;b), it has never been used on its own without any need for STE or other quantization methods, while achieving state-of-the-art performance.

**Our Contribution:** (i) With DIFFQ, we propose to use pseudo quantization noise **only** to approximate quantization at train time, as a differentiable alternative to STE, both with respect to the unquantized weights and number of bits used.
(ii) We provide a differentiable model size estimate, so that given a single penalty level $\lambda$, DIFFQ optimizes the number of bits per weight or group of weights to achieve a given trade-off between model size and accuracy.
(iii) We provide extensive experimental validation using various models (ConvNets and Transformers) and domains (image classification, language modeling, audio source separation). We demonstrate the efficiency of DIFFQ both in providing small footprint models with comparable performance to the uncompressed ones, together with easy and stable optimization, using only one sensitive hyper-parameter.

## 2 Related Work

Early network quantization methods focused on low-bitwidth networks such as BinaryNet Courbariaux et al. (2015; 2016), XOR-Nets Rastegari et al. (2016), or Ternary networks Li et al. (2016); Wu et al. (2018). Although these methods produce highly quantized models, their performance is not on par with uncompressed ones. To improve accuracies, higher bitwidth quantization methods were studied Jung et al. (2019); Zhang et al. (2018a); Mishra et al. (2017). These methods followed the STE approach Bengio et al. (2013). STE allows the gradients to be backpropagated through the quantizers and, thus, the network weights can be adapted with gradient descent Courbariaux et al. (2016).

Variational approaches were used to make the categorical distribution over quantized weights differentiable. Louizos et al. (2019) uses a Gumbel-softmax (Jang et al., 2017) but requires 2 hyper-parameters and has no bitwidth tuning. DIFFQ has a single hyper-parameter and supports automatic bitwidth tuning. Shayer et al. (2018) relies on a Central Limit Theorem (CLT) application, however this prevents weights from converging to a deterministic value, which would break the assumptions of the CLT. With DIFFQ, weights are free to converge to any optimal value. Finally Ullrich et al. (2017) uses a gaussian mixture model trained on top of the weights, adding significant complexity both in terms of code, and computation. In contrast, DIFFQ adds only one penalty term to the loss, optimized along the rest of the model in an end-to-end fashion.

An alternative is to use a smoothed version of the quantization operator, possibly with a trained meta-network (Chen et al., 2019), however as the smoothed operator converges to the true one, gradients will eventually be zero almost everywhere. Gong et al. (2019) use a meta-network to provide gradients despite quantization. However, their implementation for training the meta-network still relies on STE.

Additive noise injection has been studied by Baskin et al. (2018a), although only during the first few epochs, after which STE based approximation is used. This work was extented to non uniform quantization (Baskin et al., 2018b). In contrast, DIFFQ uses only noise injection, and as demonstrated in Results Section, achieves a better accuracy for an equivalent compression level than both methods. Non uniform quantization was also studied by Polino et al. (2018), but without differentiability with respect to the weights, with worse

performance than DIFFQ. Additive noise was also studied in the context of image compression (Ballé et al., 2017; Choi et al., 2019) in order to provide a differentiable pseudo-quantization operator. However, those work rely on an explicit estimation of the quantized values entropy, in particular with respect to a distribution of images. This formalism breaks down when having to quantize a single model, not a distribution, and DIFFQ uses a simpler approach where the bitwidth is directly tuned. More recently, Park et al. (2022) extended our method for activation quantization.

An important contribution from DIFFQ is the automatic tuning of the bitwidth using mixed-precision. Other mixed-precision quantization methods are based on Reinforcement Learning (Wang et al., 2019; Elthakeb et al., 2020; Liu et al., 2021), second-order optimization (Dong et al., 2019; 2020; Yao et al., 2021), and differentiable quantization methods (Uhlich et al., 2020; Wang et al., 2020). Comparing to DIFFQ, such methods are more complex (e.g., require plenty of parameter tuning), more computationally heavy, and most importantly based on STE approximations. Wang et al. (2019); Elthakeb et al. (2019) suggested learning a bitwidth assignment policy using reinforcement learning methods. In contrast, our method select bitwidth along training, using only first order optimization. Jain et al. (2019); Esser et al. (2020), and Bhalgat et al. (2020) proposed learning the quantizer step-size or dynamic-range using STE, but do not allow to select the bitdwidth. Our experiments show that DIFFQ outperforms (Esser et al., 2020) (LSQ) both on most vision and natural language tasks. Uhlich et al. (2020) proposed a re-parametrization that allows to select the bitwidth for each layer through first order optimization, while also relying on STE. The re-parametrization is more complex than the additive noise used in DIFFQ, and suffers from the biased gradient of STE. Results suggest that DIFFQ achieves similar or better trade-offs between model size and accuracy. Besides, in the present work we explore setting a bitwidth for individual groups of weights within each layer, rather than layer-wise.

The limitations of STE methods for quantization were first noticed by Liu & Mattina (2019). They recommend using a linear combination of the unquantized and quantized weight, with the gradient flowing only through the unquantized contribution. In a similar spirit, Fan et al. (2021) sample for each layer and iteration whether to use the quantized or unquantized weight. Both methods reduce the bias from STE, but also remove some of the quantization noise during training. In contrast our method allows to keep a full pseudo quantization noise without the STE bias. Liu et al. (2022) proposed the *Generalized STE* method to deal with gradient instabilities by calculating the expectation of the stochastic quantization during the backward phase. Finally, Nagel et al. (2022) extend the analysis we present in Section 3.3 on the oscillations of weights when using STE and suggest tracking the weight oscillations in order to freeze them when needed, as an ad-hoc solution.

A last line of related work is Product Quantization (PQ) Stock et al. (2019), where code words are being learned to quantize blocks of weights rather than single weights. This method achieves a higher compression level than per-weight quantization but also requires carefully choosing the size of the codebooks for each layer. In contrast, our method requires only choosing a single hyper-parameter to balance between model size and accuracy. Besides, as noted by Fan et al. (2021), per-weight quantization and PQ can be combined. We compare with PQ on vision and language tasks: while PQ can reach smaller model size than DIFFQ, it can also suffer from unacceptable accuracy loss, in particular for language modeling.

## 3    Background

Let us consider a weight vector $w \in \mathbb{R}^d$, where $d \in \mathbb{N}$, typically the weights of convolution or linear layer. Each entry of the vector is typically coded over 32 bits with floating-point precision. We aim to reduce the number of possible states to $2^B$, where $B \ll 32$ is the number of bits of precision. First, we assume $w_i \in [0, 1]$ for all $1 \leq i \leq d$. In practice, one would first normalize $w$ as

$$\hat{w} = \frac{w - \min(w)}{\max(w) - \min(w)},$$

and provide the tuple $(\min(w), \max(w))$ separately as a 32 bits IEEE float. Given that for typical deep learning models $d \gg 1$, storing this range has a negligible cost. For readability, we describe the method for scalar values $w \in [0, 1]$, however, this can be easily extended to vectors $w \in \mathbb{R}^d$.

### 3.1 Uniform quantization

The simplest quantization methods consist of taking $2^B$ points evenly spaced in the range $[0, 1]$ and round each entry of $w$ to the nearest point. One can then store the rounded value by its index, which requires only $B$ bits. Formally, we quantize a number $w \in [0, 1]$ over $B$ bits as

$$\forall w \in [0, 1], B \in \mathbb{N}_*, \mathbf{Q}(w, B) = \frac{\text{round}\left(w \cdot (2^B - 1)\right)}{2^B - 1}. \tag{1}$$

While the intuitive definition of quantization is for an integer number of bits, we can extend the previous definitions to a real-valued number of bits $B \in \mathbb{R}_{*+}$. Note that variants of this scheme exist, for instance, symmetric uniform quantization, which enforces that 0 is always exactly represented (Krishnamoorthi, 2018).

### 3.2 Optimization of the quantized weights

The weight vector $w$ is typically obtained through the process of training a predictor function parameterized by $w$, denoted as $f_w$, to minimize a loss function $L$,

$$\min_{w \in \mathbb{R}^d} L(f_w), \tag{2}$$

where $L(f_w)$ is the empirical risk over a given dataset. The process of quantizing a vector $w$ over $B$ bits introduces a quantization noise $\mathbf{N}(w, B) = \mathbf{Q}(w, B) - w$, which is unaware of the training objective $L$. Even if $w$ is close to the optimum, $\mathbf{Q}(w, B)$ might deteriorate arbitrarily the performance of the predictor.

Thus, given a fixed budget of bits $B$, one would ideally like to minimize the empirical risk when considering the quantization process,

$$\min_{w \in \mathbb{R}^d} L(f_{\mathbf{Q}(w, B)}), \tag{3}$$

where $f_{\mathbf{Q}(w, B)}$ is the predictor function using the quantized model parameters.

Unfortunately, the gradients of $\mathbf{Q}(w, B)$ are zero over its definition domain because of the rounding operation, and as a result, it cannot be optimized using first-order optimization methods such as SGD or Adam (Kingma & Ba, 2015). One possible solution is to replace the Jacobian of $\mathbf{Q}(\cdot, B)$ with the identity matrix during the backward phase, as suggested in the STE method (Bengio et al., 2013). The STE method was popularized for quantization as the Quantization Aware Training (QAT) technique by Krishnamoorthi (2018).

### 3.3 The instability and bias in STE

As described by Fan et al. (2021), following the STE approach can cause instability during training and bias in the models' gradients and weights. As a result optimization will fail to converge to the optimal value even on simple cases. To demonstrate that, consider the following 1D least-mean-square problem, where $B \in \mathbb{N}_*$, the optimal weight $w_* \in [0, 1]$ such that $\mathbf{Q}(w_*, B) \neq w_*$, and $\mathbf{Q}(w_*, B) \in (0, 1)$. Given a random variable $X \in \mathbb{R}$ with $\sigma^2 = \mathbb{E}\left[X^2\right]$ such that $0 < \sigma^2 < \infty$, we would like to minimize the following using STE based QAT:

$$\min_{w \in [0, 1]} L(w) := \mathbb{E}\left[\frac{1}{2}\left(X\mathbf{Q}(w, B) - Xw_*\right)^2\right]. \tag{4}$$

We immediately have that the optimum is achieved for $\mathbf{Q}(w, B) = \mathbf{Q}(w_*, B)$. Let us try to optimize equation 4 using SGD with STE starting from $w_0 = w_*$, with $w_n$ the sequence of iterates. We call $w_-$ and $w_+$ the quantized values just under and above $w_*$, and we assume without loss of generality that $\mathbf{Q}(w_*, B) = w_+$. The expected gradient with STE at iteration $n$ is given by

$$G_n = \sigma^2(\mathbf{Q}(w_n, B) - w_*). \tag{5}$$

In particular, $G_0 = \sigma^2(w_+ - w_*) > 0$, and $G_n$ will stay positive until $\mathbf{Q}(w_n, B) = w_-$. At this point, we will have $G_n < 0$, and will stay so until $\mathbf{Q}(w_n, B) = w_+$. Thus, we observe that using STE, $\mathbf{Q}(w_n, B)$ will oscillate between $w_-$ and $w_+$, while the optimal value is $w_+$. The pattern of oscillation will depend on the learning

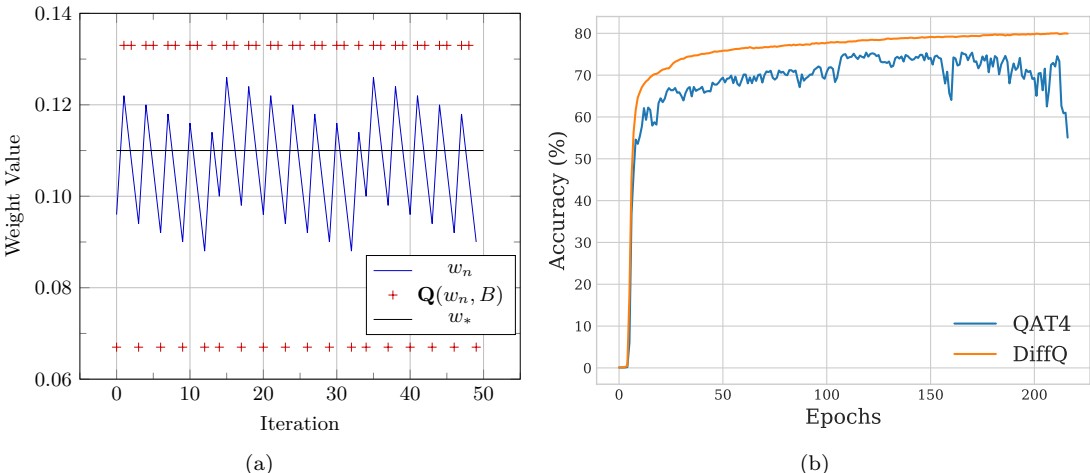

(a)                                          (b)

Figure 1: **(a)** Using STE and SGD to optimize the 1D least-mean-square problem given by equation 4 (with $B = 4$ and $X = 1$ a.s.). $\mathbf{Q}(w_n, B)$ oscillates between the quantized value just above $(w_+)$ and just under $(w_-)$ the unquantized ground truth $w_*$, while $w_n$ oscillates around the boundary $(w_+ + w_-)/2$. **(b)** Model accuracy vs. epochs for ImageNet using EfficientNet-b3. Results are presented for both QAT over 4 bits and DIFFQ.

rate and relative position of $w_*$ within the segment $[w_-, w_+]$. Taking a smaller step size will reduce the amplitude of the oscillations of $w_n$, but not of $\mathbf{Q}(w_n, B)$, which is what interests us. Indeed, $w_n$ oscillations are centered at the boundary $(w_+ + w_-)/2$. We provide one example of those oscillations on Figure 1 with $w_* = 0.11$, $B = 4$, $X = 1$ a.s. and a step size of 0.5.

Extrapolating to a model with millions of parameters, at any point in time, a significant fraction of the weights could be quantized to a suboptimal value due to the oscillations implied by the STE method. We conjecture that this behavior explains the oscillations of the accuracy observed when training an EfficientNet-b3 with QAT using 4 bits per weight on ImageNet (see Figure 1(b)). In the following section, we introduce DIFFQ, a method based on independent additive pseudo quantization noise, that does not suffer from such a bias, while approximating well enough quantization noise to perform efficient quantization aware training.

## 4    Method

**Pseudo quantization noise.** A classical assumption in digital signal processing when working with quantized signals is that the quantization noise is approximated by independents uniform variables over $[-\Delta/2, \Delta/2]$ with $\Delta = \frac{1}{2^B - 1}$ the quantization step. This approximation was studied in depth by Widrow et al. (1996) as Pseudo Quantization Noise (PQN). Following this assumption, we define the *pseudo quantization function* $\mathbf{Q}$ for all $x \in \mathbb{R}$ and $B \in \mathbb{R}_{+*}$ as

$$\mathbf{Q}(x, B) = x + \frac{\Delta}{2} \cdot \mathcal{U}[-1, 1], \tag{6}$$

with $\mathcal{U}[-1, 1]$ an independent sample from the uniform distribution over $[-1, 1]$. This pseudo quantization function is differentiable with respect to $x$ and $B$. Unlike QAT, this differentiability does not require an STE. It also provides a meaningful gradient with respect to the number of bits used $B$ (extended to be real-valued).

If we look back at the example from Figure 1, using now equation 6 instead of STE, the expected gradients for SGD become

$$\begin{aligned} G_n &= \mathbb{E}\left[x \cdot \left(\left(w_n + \frac{\Delta}{2} \cdot \mathcal{U}[-1, 1]\right)x - w_* x\right)\right] \\ &= \sigma^2(w_n - w_*), \end{aligned} \tag{7}$$

which cancels out for $w_n = w_*$, so that at convergence we indeed have $\mathbf{Q}(w_n, B) = \mathbf{Q}(w_*, B)$, i.e. the gradient estimate is unbiased and converges to the right solution.

**Mixed precision.**   We used a common precision $B$ for all the entries of the weight vector $w$. One can instead use different values for different entries. Formally, the entries in $w$ are grouped by considering $w \in \mathbb{R}^{g \times d/g}$ with $g$ the group size and $d/g$ the number of groups. We can then extend the definition of $\mathbf{Q}(w, B)$ given by equation 1 and equation 6 to use a number of bits $b_s$ for the group $s$, with $b \in \mathbb{R}*+^{d/g}$.

**Training objective.**   Given $w \in \mathbb{R}^{g \times d/g}$ with $g$ groups of $d/g$ entries, and a number of bits $b \in \mathbb{N}_*^g$, we define the model size, expressed in MegaBytes ($1\text{MB} = 8 \cdot 2^{20}$ bits)

$$\mathbf{M}(b) = \frac{g}{2^{23}} \sum_{s=1}^{d/g} b_s. \tag{8}$$

A typical objective of quantization is to achieve the best possible performance within a given model size budget or to achieve the smallest model size that reaches a given performance, i.e. we want to minimize with $b \in \mathbb{N}_*^{d/g}$, and $w \in \mathbb{R}^{g \times d/g}$ either,

$$\begin{aligned} \min_{w,b} L(f_{\mathbf{Q}(w,b)}), \qquad && \min_{w,b} \mathbf{M}(b), \\ \text{s.t.} \quad \mathbf{M}(b) \le m. \qquad \text{or} \qquad && \text{s.t.} \quad L(f_{\mathbf{Q}(w,b)}) \le l. \end{aligned} \tag{9}$$

We can relax $b$ to be real valued, and replace $\mathbf{Q}$ by our differentiable pseudo quantization function $\mathbf{Q}$. Then, following the *exact penalty method* (Bertsekas (1997), Section 4.2, Bertsekas (2014), Chapter 4), there is $\lambda(m) > 0$ (or $\lambda(l)$ for the right hand side problem), such that the left hand size problem is equivalent to

$$\min_{w,b} L(f_{\mathbf{Q}(w,b)}) + \lambda(m)\mathbf{M}(b), \tag{10}$$

which is fully differentiable with respect to $w$ and $b$ and can be optimized with first order optimization.

**Parametrization.**   In practice, the number of bits used for each group $b \in \mathbb{R}_{*+}^g$ is obtained from a logit parameter $l \in \mathbb{R}^g$, so that we have

$$b = b_{\min} + \sigma(l)(b_{\max} - b_{\min}), \tag{11}$$

with $\sigma$ is the sigmoid function, and $b_{\min}$ and $b_{\max}$ the minimal and maximal number of bits to use. The trainable parameter $l$ is initialized so that $b = b_{\text{init}}$. We set $b_{\text{init}} = 8$.

**Evaluation and noise distribution.**   At evaluation time, we round the value $b$ obtained from equation 10 as $\tilde{b} = \text{round}(b)$ and quantize $w$ as $\mathbf{Q}(w, \tilde{b})$. Thus, the amount of quantization noise at evaluation can be larger than the amount of noise injected at train time. We observed that using a noise distribution with larger support, such as Gaussian noise with unit variance (i.e. 3 times the variance of $\mathcal{U}([-1, 1])$), makes the model more robust to this operation. An empirical comparison between uniform and Gaussian noise can be found in Table B.7 in the Appendix. Thus in the rest of the paper, we always use Gaussian noise at train time.

**True model size.**   The mode size given by equation 8 is used at train time but does not account for part of the true model size. At evaluation time, we represent each weight by the integer obtained from the rounding operation in equation 1. For each layer in the network, we also store two 32 bits float numbers for the minimum and maximum scale. Finally, the actual value of $\tilde{b}$ must be coded, as it is no longer a fixed constant. For each layer, we compute the maximum value of $C_s = \log_2(1 + \tilde{b}_s - b_{\min})$ over all groups $s \in \{1, \ldots, d/g\}$. We encode once the value $\max(C)$ as an 8-bit integer, and for each group, we encode $b_s - b_{\min}$ over $\max(C)$ bits. The true size for one layer, expressed in MegaBytes, is thus given by

$$\tilde{\mathbf{M}}(b) = \frac{1}{2^{23}}\left(2 \cdot 32 + 8 + \frac{d}{g}\max(C) + g\sum_{s=1}^{d/g} b_s\right). \tag{12}$$

Table 1: Comparison of DIFFQ against baselines presented in the Related Work section. Sizes marked with $^\dagger$ are reported after Huffman coding, following Polino et al. (2018). Accuracies marked with $^*$ are the best rather than last one to match previous practices.

| MODEL | METHOD | TOP-1 ACC. (%) | M.S. (MB) |
|---|---|---|---|
| | CIFAR10 | | |
| RESNET-18 | UNCOMPRESSED | **95.3** | 42.7 |
| RESNET-18 | UNIQ BASKIN ET AL. (2018B) | 89.1 | **2.7** |
| RESNET-18 | NICE BASKIN ET AL. (2018A) | 92.7 | **2.7** |
| RESNET-18 | DIFFQ (OURS) | **93.9** | **2.7** |
| RESNET-20 | UNCOMPRESSED | **92.7**$^*$ | 1.48 |
| RESNET-20 | DQ UHLICH ET AL. (2020) | 91.4$^*$ | 0.07 |
| RESNET-20 | DIFFQ (OURS) | **91.6**$^*$ | **0.06** |
| | CIFAR100 | | |
| WIDE-RESNET | UNCOMPRESSED | **76.2** | 139.4 |
| WIDE-RESNET | DIFFQUANT POLINO ET AL. (2018) | 49.3 | 7.9 |
| WIDE-RESNET | DIFFQ (OURS) | **75.6** | **4.7** |
| | IMAGENET | | |
| RESNET-18 | UNCOMPRESSED | 70.9$^*$ | 44.6 |
| RESNET-18 | META-QUANT CHEN ET AL. (2019) | 60.3 | **1.3** |
| RESNET-18 | DQ UHLICH ET AL. (2020) | 70.1$^*$ | 5.4 |
| RESNET-18 | LSQ 4 BITS ESSER ET AL. (2020) | 70.7$^*$ | 5.6 |
| RESNET-18 | DIFFQ (OURS) | **71.1**$^*$ | **5.3** |
| RESNET-50 | UNCOMPRESSED | **77.1**$^*$ | 97.5 |
| RESNET-50 | LSQ 4 BITS ESSER ET AL. (2020) | 76.2$^*$ | 12.3 |
| RESNET-50 | LSQ 3 BITS ESSER ET AL. (2020) | 75.6$^*$ | 9.3 |
| RESNET-50 | DIFFQ (OURS) | **76.6**$^*$ | 10.5 |
| RESNET-50 | DIFFQ (OURS) | 76.3$^*$ | **8.8** |

## 5 Results

We present experimental results for language modeling, audio source separation, and image classification. We show that DIFFQ can often provide a model with comparable performance to the uncompressed one while producing a model with a smaller footprint than the baseline methods (STE based). We provide a finer analysis of different aspects of DIFFQ hyper-parameters and their impact on quantized models in next Section. Finally, we discuss limitations of DiffQ in the Limitation Section. Both experimental code, and a generic framework usable with any architecture in just a few lines, is available on our Github github.com/facebookresearch/diffq. All hyper-parameters for optimization and model definition are detailed in the Appendix. In all tables, ↑ (resp. ↓) indicates that highest is best (resp. lowest is best). All results referred to as "QAT" are obtained using the formula given by equation 1 with a layer-wise min-max scaling of the weights. When using DIFFQ, we use the same per layer min-max scaling. When also doing activation quantization, we use per-channel min-max scaling of the activations. All DIFFQ experiments use Gaussian noise as explained in Section 4.

### 5.1 Comparison to related work

On Table 1, we compare DIFFQ to some of the related work presented in Section 2. Compared with the NICE (Baskin et al., 2018a) and UNIQ (Baskin et al., 2018b) methods, which also rely on additive noise, DIFFQ achieves significantly better accuracy for the same model size. We then compare to the differentiable

Table 2: Language modeling results for a 16 layer Transformer trained on Wikitext-103. We also test combining weight and activation quantization. We compared DIFFQ to QAT and Quant-Noise (QN) method proposed by Fan et al. (2021) (models with † were trained with a layer-drop of 0.2 Fan et al. (2019)). Activations are quantized over 8 bits, with a per-channel scaling.

| WEIGHTS | ACTIVATION | PPL ↓ | M. S. (MB) ↓ |
|---|---|---|---|
| UNCOMPRESSED | - | **18.1** | 942 |
| 8 BITS | 8 BITS | 18.3 | 236 |
| QAT 8BITS | 8 BITS | 19.7 | 236 |
| QAT 4BITS | 8 BITS | 29.9 | 118 |
| LSQ 4 BITS (ESSER ET AL., 2020) | 8 BITS | 18.9 | 118 |
| DIFFQ ($\lambda{=}5, g{=}16$) | 8 BITS | **18.1** | 130 |
| DIFFQ ($\lambda{=}10, g{=}16$) | 8 BITS | 18.6 | **113** |
| UNCOMPRESSED † | - | 18.3 | 942 |
| QN 8 BITS† FAN ET AL. (2021) | QN 8 BITS | 18.7 | 236 |
| QN 4 BITS† FAN ET AL. (2021) | QN 8 BITS | 19.5 | 118 |
| PQ† FAN ET AL. (2021) | - | 20.7 | **38** |

quantization method by (Polino et al., 2018), which only optimizes the non uniform quantization points, not the pre-quantization weights. Following their practice, we report numbers after Huffman coding. We achieve a model almost half as small, with a gap of 25% in accuracy, proving that optimizing pre-quantization weights is more important than tuning a non uniform quantization grid. Meta-Quant (Chen et al., 2019) achieves smaller model size than DIFFQ, with 1 bit per weight, a regime where the PQN assumption breaks down, at the price of losing nearly 10% of accuracy. Finally, compared with two quantization methods: DQ by Uhlich et al. (2020) and LSQ by Esser et al. (2020). When considering DQ, DIFFQ achieves slightly smaller model size and better accuracy on ImageNet using ResNet-18, and a 15% smaller model with sightly better accuracy for a Resnet-20 trained on CIFAR-10. Comparing to LSQ [1], DIFFQ achieves better accuracy with smaller model size on ImageNet using both ResNet-18 and ResNet-50. Additional comparison between DIFFQ and LSQ for higher compression rates can be on Table B.1 in the Appendix.

## 5.2 Language Modeling

We trained a 16 layers transformer (Vaswani et al., 2017) based language model on the Wikitext-103 text corpus (Merity et al., 2016), following Baevski & Auli (2019), and using the Fairseq framework (Ott et al., 2019). Results are presented in Table 2. We compare to the Quant-Noise method by Fan et al. (2021), but use a reduced layer-drop (Fan et al., 2019) of 0.1 instead of 0.2. This both improves the baseline, as well as the performance of DIFFQ models. For DIFFQ, we explicitly set the gradient for the number of bits parameters to zero for all layers that have been dropped. In order to test the compatibility of DIFFQ with efficient int8 kernels, we further quantize the activations to 8 bits using PyTorch native support (Paszke et al., 2019).

The transformer model has some tied parameters (e.g. word embedding in the first and pre-softmax layer). It is important to detect such tied parameters with DIFFQ. We use a single shared bits parameter when a parameter tensor is reused multiple times, and for each forward, we sample a single pseudo quantization noise per group of shared weights and reuse it appropriately. Failure to do so led to a significant worsening of the performance at validation time.

While QAT breaks down when trying to get to 4 bits precision (perplexity of 29.9), using DIFFQ allows to achieve a lower model size (113MB vs. 118 MB for QAT 4 bits) with a perplexity closer to the uncompressed one (18.6, vs. 18.1 uncompressed). We also tried fine-tuning a pre-trained model with LSQ (Esser et al., 2020). While this works better than QAT, LSQ reaches a worst perplexity for a slightly larger model size

---

[1] We used our own LSQ implementation, with only weight quantization, since no official code is available. Comparison with the results reported in Esser et al. (2020) can be found on Table B.1.

Table 3: Music source separation results for the Demucs model (Défossez et al., 2019). We report Signal-to-Distortion Ration (SDR) together with Model Size (M.S.).

|  | SDR (dB) ↑ | M. S. (MB) ↓ |
|---|---|---|
| UNCOMPRESSED | 6.31 | 1014 |
| QAT 4BITS | 5.99 | 130 |
| QAT 5BITS | **6.27** | 162 |
| DIFFQ ($\lambda$=3e−4) | **6.28** | **120** |

than DIFFQ (18.9 perplexity for 118 MB). Similarly, Quant-Noise (Fan et al., 2021) improves on QAT but performs worse than DIFFQ, even when using more than twice as many bits. With just 4.4 bits per weight on average, DIFFQ achieve the same perplexity as the baseline. We also compare to PQ (Stock et al., 2019), as reported by Fan et al. (2021). While PQ achieves higher compression levels, with just 38MB, its perplexity is the worst of all methods.

### 5.3 Music Source Separation

We use the Demucs architecture by Défossez et al. (2019) with 64 initial hidden channels. The model is trained on the standard MusDB benchmark (Rafii et al., 2017), for 180 epochs, and evaluated with the Signal-To-Distortion Ratio (SDR) metric (Vincent et al., 2006). The unquantized model is 1GB. We compare DIFFQ with QAT training with either 5 or 4 bits, with the results presented in Table 3. With 5 bits, QAT is able to replicate almost the same performance as the uncompressed model. When trying to further compress the model to 4 bits per weight, QAT leads to a sharp decrease of the SDR, losing 0.3dB, for a 130MB model. DIFFQ achieves a model size of 120MB, with only a drop of 0.03dB of SDR compared to the uncompressed baseline.

### 5.4 Image Classification

Next, we evaluated three image classification benchmarks: ImageNet Deng et al. (2009), CIFAR-10 and CIFAR-100 Krizhevsky et al. (2009). For CIFAR-10 and CIFAR-100 results are reported for MobileNet-v1 Howard et al. (2017), ResNet-18 He et al. (2016), and Wide-ResNet with 28x10, depth and width levels respectively Zagoruyko & Komodakis (2016). ImageNet results are reported using EfficientNet-B3 Tan & Le (2019) and DeiT-B Touvron et al. (2020) models. More details regarding hyper-parameters and augmentations used can be found in the Appendix.

**CIFAR10 & CIFAR-100.** Results for CIFAR10 and CIFAR100 are depicted in Figures 2(a) and 2(b). We compare DIFFQ, QAT and LSQ (without activation quantization) using 2, 3, and 4 bits quantization. Performance of the uncompressed model is additionally presented as an upper-bound. To better understand the effect of the penalty level $\lambda$ on both model size and accuracy, we train models with DIFFQ using different penalty levels. Exact results are presented in Table B.2, in the Appendix, together with a detailed analysis.

Results suggest DIFFQ models reach comparable performance to the LSQ and outperforms QAT models while producing models with a smaller footprint. When considering 2 bits quantization, QAT is always worse than both LSQ and DIFFQ. While LSQ works well for Resnet18, it suffers from large drops in accuracies for MobileNet and WideResNet, failing entirely to train for MobileNet on CIFAR10, despite initialization from a pre-trained model.

**ImageNet - DeiT.** Results for ImageNet using DeiT-B model are presented in Table 4. We compared DIFFQ to QAT when training with 4 and 8 bits. Both QAT with 8 bits and DIFFQ reach comparable performance to the uncompressed model, while DIFFQ yields a model almost half of the size as QAT, however still bigger than QAT with 4 bits. When we increase $\lambda$, we get a smaller model-size than QAT with 4 bits but with better accuracy levels.

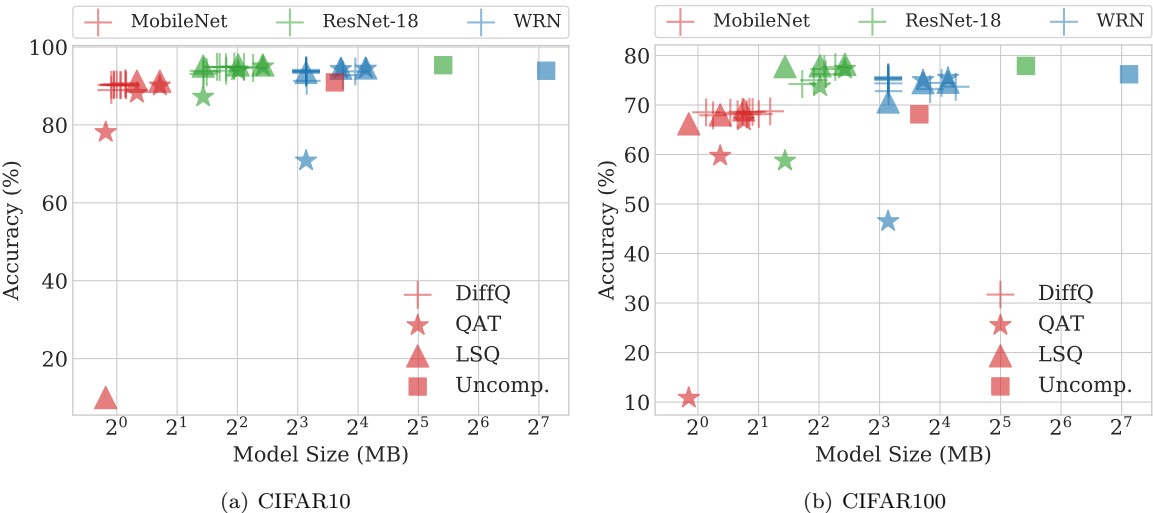

Figure 2: Model accuracy and size on CIFAR10 **(a)** and CIFAR100 **(b)** using MobileNet, ResNet-18, and WideResNet (WRN) models for various penalty levels using DIFFQ, QAT, LSQ, and the baseline.

Table 4: Image classification results for the ImageNet benchmark. Results are presented for DIFFQ and QAT using 4 and 8 bits using the DeiT model (Touvron et al., 2020). We report Top-1 Accuracy (Acc.) together with Model Size (M.S.).

|  | TOP-1 ACC. (%) ↑ | M.S. (MB) ↓ |
|---|---|---|
| UNCOMPRESSED | 81.8 | 371.4 |
| QAT 4BITS | 79.2 | 41.7 |
| QAT 8BITS | 81.6 | 82.9 |
| DIFFQ ($\lambda$=1e−2) | **82.0** | 45.7 |
| DIFFQ ($\lambda$=0.1) | 81.5 | **33.02** |

**ImageNet - EfficientNet.** We evaluate the performance of DIFFQ on the memory-efficient EfficientNet-B3 model. Results are depicted on Figure B.1 (c) as well as in Table B.5, both in the Appendix. Both QAT 8 bits and DIFFQ achieves similar accuracy (QAT 81.3 %, DIFFQ 81.5%) but with a smaller model size for DIFFQ (8.5MB vs. 12MB for QAT). When considering QAT 4 bits, DIFFQ produces a smaller model with a significantly better accuracy level (80.8%). For QAT 4, we noticed considerable instability close to the end of the training, see Figure B.1 (b) in the Appendix.

## 5.5 Analysis

**Bits Histogram.** Figure 3 presents the weight bitwidth assignment over layer groups for the EfficientNet-B3 Tan & Le (2019) and DeiT Touvron et al. (2020) models trained on ImageNet. The capacity distribution over depth for ConvNets (EfficientNet-B3) and Transformers (DeiT) are different (fp32 shows uncompressed capacity). Notice, that the quantization trends are different too: for the ConvNet, smaller bitwidths are used for deeper layers of the model while large bitwidth is more common in the first layers (except for the last linear layer which seems to need some precision). For the Transformer, this effect of varying quantization by layer is similar but less pronounced, due to the more symmetric nature of the architecture.

**Fixed bitwidth.** On Table B.4 in the Appendix, we compare QAT to DIFFQ using a fixed number of bits, i.e. comparing strictly PQN to STE. On MobileNet, ResNet-18, and WideResNet for both CIFAR10 and

CIFAR100, DIFFQ outperforms QAT, with a gap especially noticeable for 2 bits models, a regime where QAT becomes unstable, as we described in previous section.

**Group size.** We additionally evaluate the affect of the group-size, $g$, on model size and accuracy, by optimizing DIFFQ models using $g \in \{1, 4, 8, \infty\}$. When $g=\infty$, we use a single group for the entire layer. Results for ResNet-18 using CIFAR-100 are depicted in Figure 1(a) in the Appendix. Interestingly, we observed that increasing $g$, yields in a smaller model size on the expense of a minor decrease in performance. However, when setting $g=\infty$ model performance (model size and accuracy) is comparable to $g=8$ for this task.

**Runtime overhead and loading time.** Using DiffQ usually increase the training time by some amount. On the language modeling task, the time per batch went from 115ms to 125ms. When training a ResNet18 on CIFAR-10, it increased from 120ms to 150ms. For the Demucs model, it went from 0.9s to 1.1s. However, when training the EfficientNet-b3 model, we observed that the time per batch would nearly double. Thus it seems that for most architectures the training time overhead is limited, although the worst case can be up to twice as slow. At evaluation time, decompressing the Demucs model from its variable bitwidth compact representation takes around 2.81 seconds on a MacBook Pro with 2.4 GHz 8 cores Intel i9 processor.

### 5.6 Limitations

The model size given by equation 12 is obtained with a traditional encoding of the quantized model. However, more efficient coding techniques exist when the entropy of the data is low, such as Huffman coding (Huffman, 1952). Using the ZLib library, we obtain an estimate of the Huffman compressed model size after quantization. For instance, for the language model described in Table 2, the QAT 8 model gets further compressed from 236MB to 150MB, showing that the entropy of its quantized weight is significantly lower than the maximal one for 8 bits integers. However, the DIFFQ model naive size is 113MB, and after compression by ZLib, gets to 122MB. This is a sign that the entropy is close to its maximal value, with ZLib adding only overhead for no gain. In equation 10, we only penalize the naive number of bits used, while asking for the best possible accuracy. In that case, the model maximally use the entropy capabilities for a given number of bits. An interesting line of research would be to replace the model size equation 8 to account for the actual entropy of the data, for instance with differentiable kernel density estimation. We leave that for further research.

Another limitation of DiffQ is that it can make training up to twice as slow, due to the extra parameters to optimize for and the more complex gradient calculation graph. Besides, in order to achieve a specific model size or accuracy, one has to tune the $\lambda$ penalty parameter.

## 6 Discussion

We presented DIFFQ, a novel and simple differentiable method for model quantization via pseudo quantization noise addition to models' parameters. Given a single hyper-parameter that quantifies the desired trade-off between model size and accuracy, DIFFQ can optimize the number of bits used for each trainable parameter or group of parameters during model training. We conduct expensive experimental evaluations on various domains using different model architectures. Results suggest that DIFFQ is superior to the baseline methods on several benchmarks from various domains. On ImageNet, Wikitext-103, and MusDB, we achieve a model size that is smaller than a 4 bits quantized model, while retaining the same performance as the unquantized baseline. For future work, we consider adapting the model size penalty to account for Huffman encoding, which could allow to further reduce the model size when it is gzipped. Another line of work would be using PQN to improve activation quantization, enabling 4-bits kernels for a larger number of tasks.

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

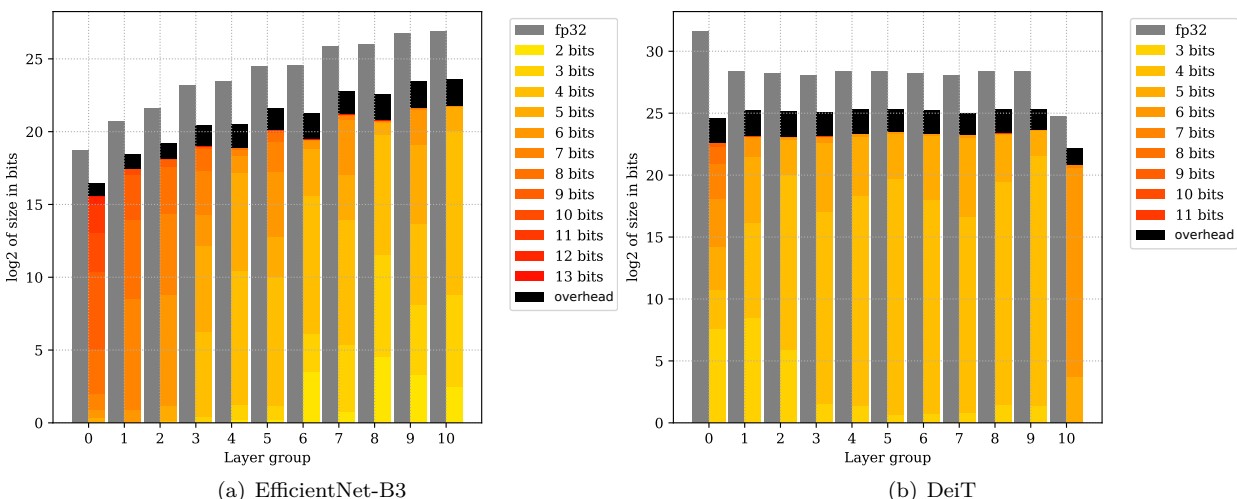

(a) EfficientNet-B3      (b) DeiT

Figure 3: We group layers of a given architecture into 11 groups (group 0 being closest to the input, and 10 closest to the output), and report for each group its contribution to the model size. We compare the baseline EfficientNet-B3 (above) and DeiT (below) models (floating point 32 bits) and the quantized models with DIFFQ ($\lambda$=5e−3 for EfficientNet-B3, $\lambda$=1e−2 for DeiT). For quantized model, we also report the distribution over each bitwidth within each group of layers. Scale is logarithmic across layers, and linear inside each one. Finally, "overhead" shows the capacity needed to encode the bitwidth used for each group of weights.

Chaim Baskin, Natan Liss, Yoav Chai, Evgenii Zheltonozhskii, Eli Schwartz, Raja Giryes, Avi Mendelson, and Alexander M Bronstein. Nice: Noise injection and clamping estimation for neural network quantization. *arXiv preprint arXiv:1810.00162*, 2018a.

Chaim Baskin, Eli Schwartz, Evgenii Zheltonozhskii, Natan Liss, Raja Giryes, Alex M Bronstein, and Avi Mendelson. Uniq: Uniform noise injection for non-uniform quantization of neural networks. *arXiv preprint arXiv:1804.10969*, 2018b.

Yoshua Bengio, Nicholas Léonard, and Aaron Courville. Estimating or propagating gradients through stochastic neurons for conditional computation. *arXiv preprint arXiv:1308.3432*, 2013.

Dimitri P Bertsekas. Nonlinear programming. *Journal of the Operational Research Society*, 48(3):334–334, 1997.

Dimitri P Bertsekas. *Constrained optimization and Lagrange multiplier methods*. Academic press, 2014.

Yash Bhalgat, Jinwon Lee, Markus Nagel, Tijmen Blankevoort, and Nojun Kwak. Lsq+: Improving low-bit quantization through learnable offsets and better initialization. In *Proceedings of the IEEE/CVF Conference on Computer Vision and Pattern Recognition Workshops*, pp. 696–697, 2020.

Shangyu Chen et al. Metaquant: Learning to quantize by learning to penetrate non-differentiable quantization. In *Advances in Neural Information Processing Systems*, 2019.

Yoojin Choi, Mostafa El-Khamy, and Jungwon Lee. Variable rate deep image compression with a conditional autoencoder. In *Proceedings of the IEEE/CVF International Conference on Computer Vision*, pp. 3146–3154, 2019.

Matthieu Courbariaux, Yoshua Bengio, and Jean-Pierre David. Binaryconnect: Training deep neural networks with binary weights during propagations. *arXiv preprint arXiv:1511.00363*, 2015.

Matthieu Courbariaux, Itay Hubara, Daniel Soudry, Ran El-Yaniv, and Yoshua Bengio. Binarized neural networks: Training deep neural networks with weights and activations constrained to +1 or -1. *arXiv preprint arXiv:1602.02830*, 2016.

Alexandre Défossez, Nicolas Usunier, Léon Bottou, and Francis Bach. Music source separation in the waveform domain. *arXiv preprint arXiv:1911.13254*, 2019.

J. Deng, W. Dong, R. Socher, L.-J. Li, K. Li, and L. Fei-Fei. ImageNet: A Large-Scale Hierarchical Image Database. In *CVPR*, 2009.

Zhen Dong, Zhewei Yao, Amir Gholami, Michael W Mahoney, and Kurt Keutzer. Hawq: Hessian aware quantization of neural networks with mixed-precision. In *Proceedings of the IEEE/CVF International Conference on Computer Vision*, pp. 293–302, 2019.

Zhen Dong, Zhewei Yao, Daiyaan Arfeen, Amir Gholami, Michael W Mahoney, and Kurt Keutzer. Hawq-v2: Hessian aware trace-weighted quantization of neural networks. *Advances in neural information processing systems*, 33:18518–18529, 2020.

Ahmed Elthakeb, Prannoy Pilligundla, FatemehSadat Mireshghallah, Amir Yazdanbakhsh, Sicuan Gao, and Hadi Esmaeilzadeh. Releq: An automatic reinforcement learning approach for deep quantization of neural networks. In *NeurIPS ML for Systems workshop, 2018*, 2019.

Ahmed T Elthakeb, Prannoy Pilligundla, Fatemehsadat Mireshghallah, Amir Yazdanbakhsh, and Hadi Esmaeilzadeh. Releq: A reinforcement learning approach for automatic deep quantization of neural networks. *IEEE micro*, 40(5):37–45, 2020.

Steven K Esser, Jeffrey L McKinstry, Deepika Bablani, Rathinakumar Appuswamy, and Dharmendra S Modha. Learned step size quantization. In *Proc. of the International Conference on Learning Representations*, 2020.

Angela Fan, Edouard Grave, and Armand Joulin. Reducing transformer depth on demand with structured dropout. In *Proc. of the International Conference on Learning Representations*, 2019.

Angela Fan, Pierre Stock, Benjamin Graham, Edouard Grave, Remi Gribonval, Herve Jegou, and Armand Joulin. Training with quantization noise for extreme model compression. In *ICLR 2021*, 2021.

Ruihao Gong, Xianglong Liu, Shenghu Jiang, Tianxiang Li, Peng Hu, Jiazhen Lin, Fengwei Yu, and Junjie Yan. Differentiable soft quantization: Bridging full-precision and low-bit neural networks. In *Proceedings of the IEEE International Conference on Computer Vision*, 2019.

Kaiming He, Xiangyu Zhang, Shaoqing Ren, and Jian Sun. Deep residual learning for image recognition. In *Proceedings of the IEEE conference on computer vision and pattern recognition*, 2016.

Andrew G Howard, Menglong Zhu, Bo Chen, Dmitry Kalenichenko, Weijun Wang, Tobias Weyand, Marco Andreetto, and Hartwig Adam. Mobilenets: Efficient convolutional neural networks for mobile vision applications. *arXiv preprint arXiv:1704.04861*, 2017.

D. A. Huffman. A method for the construction of minimum-redundancy codes. *Proceedings of the IRE*, 40(9): 1098–1101, 1952. doi: 10.1109/JRPROC.1952.273898.

Yerlan Idelbayev. Proper ResNet implementation for CIFAR10/CIFAR100 in PyTorch. https://github.com/akamaster/pytorch_resnet_cifar10, 2018.

Sambhav R Jain, Albert Gural, Michael Wu, and Chris H Dick. Trained quantization thresholds for accurate and efficient fixed-point inference of deep neural networks. *arXiv preprint arXiv:1903.08066*, 2019.

Eric Jang, Shixiang Gu, and Ben Poole. Categorical reparameterization with gumbel-softmax. In *Proc. of the International Conference on Learning Representations*, 2017.

Sangil Jung, Changyong Son, Seohyung Lee, Jinwoo Son, Jae-Joon Han, Youngjun Kwak, Sung Ju Hwang, and Changkyu Choi. Learning to quantize deep networks by optimizing quantization intervals with task loss. In *Proceedings of the IEEE/CVF Conference on Computer Vision and Pattern Recognition*, pp. 4350–4359, 2019.

Diederik P Kingma and Jimmy Ba. Adam: A method for stochastic optimization. In *Proc. of the International Conference on Learning Representations*, 2015.

Raghuraman Krishnamoorthi. Quantizing deep convolutional networks for efficient inference: A whitepaper. *arXiv preprint arXiv:1806.08342*, 2018.

Alex Krizhevsky, Geoffrey Hinton, et al. Learning multiple layers of features from tiny images. Technical report, University of Toronto, 2009.

Fengfu Li, Bo Zhang, and Bin Liu. Ternary weight networks. *arXiv preprint arXiv:1605.04711*, 2016.

Jing Liu, Jianfei Cai, and Bohan Zhuang. Sharpness-aware quantization for deep neural networks. *arXiv preprint arXiv:2111.12273*, 2021.

Zechun Liu, Kwang-Ting Cheng, Dong Huang, Eric P Xing, and Zhiqiang Shen. Nonuniform-to-uniform quantization: Towards accurate quantization via generalized straight-through estimation. In *Proceedings of the IEEE/CVF Conference on Computer Vision and Pattern Recognition*, pp. 4942–4952, 2022.

Zhi-Gang Liu and Matthew Mattina. Learning low-precision neural networks without straight-through estimator (ste). *arXiv preprint arXiv:1903.01061*, 2019.

Ilya Loshchilov and Frank Hutter. Decoupled weight decay regularization. In *Proc. of the International Conference on Learning Representations*, 2019.

Christos Louizos, Matthias Reisser, Tijmen Blankevoort, Efstratios Gavves, and Max Welling. Relaxed quantization for discretized neural networks. In *Proc. of the International Conference on Learning Representations*, 2019.

Stephen Merity, Caiming Xiong, James Bradbury, and Richard Socher. Pointer sentinel mixture models. In *Proc. of the International Conference on Learning Representations*, 2016.

Paulius Micikevicius, Sharan Narang, Jonah Alben, Gregory Diamos, Erich Elsen, David Garcia, Boris Ginsburg, Michael Houston, Oleksii Kuchaiev, Ganesh Venkatesh, et al. Mixed precision training. In *Proc. of the International Conference on Learning Representations*, 2018.

Asit Mishra, Eriko Nurvitadhi, Jeffrey J Cook, and Debbie Marr. Wrpn: Wide reduced-precision networks. In *Proc. of the International Conference on Learning Representations*, 2017.

Markus Nagel, Marios Fournarakis, Yelysei Bondarenko, and Tijmen Blankevoort. Overcoming oscillations in quantization-aware training. *arXiv preprint arXiv:2203.11086*, 2022.

Myle Ott, Sergey Edunov, Alexei Baevski, Angela Fan, Sam Gross, Nathan Ng, David Grangier, and Michael Auli. fairseq: A fast, extensible toolkit for sequence modeling. In *Proceedings of NAACL-HLT 2019: Demonstrations*, 2019.

Sein Park, Junhyuk So, Juncheol Shin, and Eunhyeok Park. Nipq: Noise injection pseudo quantization for automated dnn optimization. *arXiv preprint arXiv:2206.00820*, 2022.

Adam Paszke et al. Pytorch: An imperative style, high-performance deep learning library. In *Advances in Neural Information Processing Systems 32*, 2019.

Antonio Polino et al. Model compression via distillation and quantization. In *Proc. of the International Conference on Learning Representations*, 2018.

Zafar Rafii, Antoine Liutkus, Fabian-Robert Stöter, Stylianos Ioannis Mimilakis, and Rachel Bittner. The musdb18 corpus for music separation, 2017.

Mohammad Rastegari, Vicente Ordonez, Joseph Redmon, and Ali Farhadi. Xnor-net: Imagenet classification using binary convolutional neural networks. In *European conference on computer vision*, pp. 525–542. Springer, 2016.

Oran Shayer, Dan Levi, and Ethan Fetaya. Learning discrete weights using the local reparameterization trick. In *Proc. of the International Conference on Learning Representations*, 2018.

Pierre Stock, Armand Joulin, Rémi Gribonval, Benjamin Graham, and Hervé Jégou. And the bit goes down: Revisiting the quantization of neural networks. In *Proc. of the International Conference on Learning Representations*, 2019.

Christian Szegedy, Vincent Vanhoucke, Sergey Ioffe, Jon Shlens, and Zbigniew Wojna. Rethinking the inception architecture for computer vision. In *Proceedings of the IEEE conference on computer vision and pattern recognition*, pp. 2818–2826, 2016.

Mingxing Tan and Quoc V Le. Efficientnet: Rethinking model scaling for convolutional neural networks. In *Proc. of the International Conference on Machine Learning*, 2019.

Tijmen Tieleman and Geoffrey Hinton. Lecture 6.5-rmsprop: Divide the gradient by a running average of its recent magnitude, 2012.

Hugo Touvron, Matthieu Cord, Matthijs Douze, Francisco Massa, Alexandre Sablayrolles, and Hervé Jégou. Training data-efficient image transformers & distillation through attention. *arXiv preprint arXiv:2012.12877*, 2020.

Stefan Uhlich, Lukas Mauch, Fabien Cardinaux, Kazuki Yoshiyama, Javier Alonso Garcia, Stephen Tiedemann, Thomas Kemp, and Akira Nakamura. Mixed precision dnns: All you need is a good parametrization. In *Proc. of the International Conference on Learning Representations*, 2020.

Karen Ullrich, Edward Meeds, and Max Welling. Soft weight-sharing for neural network compression. In *Proc. of the International Conference on Learning Representations*, 2017.

Ashish Vaswani, Noam Shazeer, Niki Parmar, Jakob Uszkoreit, Llion Jones, Aidan N Gomez, Lukasz Kaiser, and Illia Polosukhin. Attention is all you need. In *Proc. of Neural Information Processing Systems*, 2017.

Emmanuel Vincent, Rémi Gribonval, and Cédric Févotte. Performance measurement in blind audio source separation. *IEEE Transactions on Audio, Speech and Language Processing*, 2006.

Kuan Wang et al. Haq: Hardware-aware automated quantization with mixed precision. In *Proceedings of the IEEE/CVF Conference on Computer Vision and Pattern Recognition*, pp. 8612–8620, 2019.

Ying Wang, Yadong Lu, and Tijmen Blankevoort. Differentiable joint pruning and quantization for hardware efficiency. In *European Conference on Computer Vision*, pp. 259–277. Springer, 2020.

Bernard Widrow, Istvan Kollar, and Ming-Chang Liu. Statistical theory of quantization. *IEEE Transactions on instrumentation and measurement*, 45(2):353–361, 1996.

Ross Wightman. Pytorch image models. https://github.com/rwightman/pytorch-image-models, 2019.

Shuang Wu, Guoqi Li, Feng Chen, and Luping Shi. Training and inference with integers in deep neural networks. In *Proc. of the International Conference on Learning Representations*, 2018.

Zhewei Yao, Zhen Dong, Zhangcheng Zheng, Amir Gholami, Jiali Yu, Eric Tan, Leyuan Wang, Qijing Huang, Yida Wang, Michael Mahoney, et al. Hawq-v3: Dyadic neural network quantization. In *International Conference on Machine Learning*, pp. 11875–11886. PMLR, 2021.

Sangdoo Yun, Dongyoon Han, Seong Joon Oh, Sanghyuk Chun, Junsuk Choe, and Youngjoon Yoo. Cutmix: Regularization strategy to train strong classifiers with localizable features. In *Proceedings of the IEEE/CVF International Conference on Computer Vision*, pp. 6023–6032, 2019.

Sergey Zagoruyko and Nikos Komodakis. Wide residual networks. *arXiv preprint arXiv:1605.07146*, 2016.

Dongqing Zhang, Jiaolong Yang, Dongqiangzi Ye, and Gang Hua. Lq-nets: Learned quantization for highly accurate and compact deep neural networks. In *Proceedings of the European conference on computer vision (ECCV)*, pp. 365–382, 2018a.

Hongyi Zhang, Moustapha Cisse, Yann N Dauphin, and David Lopez-Paz. mixup: Beyond empirical risk minimization. In *Proc. of the International Conference on Learning Representations*, 2018b.

