# OpenReview forum: "Differentiable Model Compression via Pseudo Quantization Noise"
_TMLR — Accepted by TMLR_

### Review · Reviewer_d8XE · 2022-07-28

**Summary Of Contributions:**

In this paper, the authors focus on quantization-aware training and propose introducing a Pseudo quantization noise formula. The noise injection method is supposed to emulate the effect of quantization. The noise form can also be combined into a framework to learn mixed precision networks.

**Broader Impact Concerns:**

Nothing worth concern.

**Requested Changes:**

The authors need to provide a theoretical justification of this method, and comprehensive comparison with existing SOTA.

**Strengths And Weaknesses:**

Pros:

+ The proposed method is easy to follow, inspired by a classical assumption in the signal processing area.

+ The authors conducted experiments on multiple benchmarks, beyond image classification.


Cons:

- Currently, the analysis is not convincing, the authors derive the method based on a historical assumption, but for neural networks, it is unclear how this assumption will affect the optimization. I expect to see some theoretical analysis on convergence.

- Is this method applicable to activation quantization? Quantization on weights is not that hard compared to activation quantization. Therefore, it makes me question the real improvements over STE. An ablation study should be conducted.

- The mixed precision on model size is not a hard problem since the former layers have very few parameters, making them easily learn high bits. However, these early layers do not bring less computation energy or latency saving than later layers. Hence, model size mixed precision is easier for high accuracy and less likely to really achieve hardware acceleration in practice.

- The paper fails to compare recent works in QAT, they claimed the state of the arts QAT methods are proposed in the year 2020. The authors are strongly recommended to compare with works like [1, 2] (see ref below).

[1] Liu Z, Cheng K T, Huang D, et al. Nonuniform-to-Uniform Quantization: Towards Accurate Quantization via Generalized Straight-Through Estimation[C]//Proceedings of the IEEE/CVF Conference on Computer Vision and Pattern Recognition. 2022: 4942-4952.

[2] Nagel M, Fournarakis M, Bondarenko Y, et al. Overcoming Oscillations in Quantization-Aware Training[J]. arXiv preprint arXiv:2203.11086, 2022.

---

### Review · Reviewer_NAyT · 2022-07-31

**Summary Of Contributions:**

In the manuscript, the authors propose DIFFQ, which applies pseudo quantization noise during training to avoid using the standard STE method.

Applications on various tasks, such as image classification, NLP, and audio source separation, are taken to validate the effectiveness of the DIFFQ method.

The experimental results show that DIFFQ can achieve comparable or better results compared to the previous works. Additionally, the authors use different and state-of-the-art neural architectures (EfficientNetB3, DeiT, etc) in their experiments, which strengthens the claim of better performance.


**Broader Impact Concerns:**

I don't have concerns about the broader impact of this manuscript.

**Requested Changes:**

1. In the manuscript the authors mention that DIFFQ is superior to STE. It would be also important to show/discuss/compare the computational cost of applying DIFFQ and standard QAT. For example, the latency per iteration, or the throughput.
2. small issue, page 11, Figure 3, "Layer groups wise size" is not very clear.
3. Overall I think the manuscript is decent, which can lead to an acceptance if the limitations mentioned in the reviews are well addressed.

**Strengths And Weaknesses:**

Strengths:
1. The manuscript is easy to follow, with clear illustrations.
2. The related work section of the manuscript is thorough.
3. Useful information is also presented in the Appendix and the supplementary codes.
4. Although DIFFQ is not the first in this research direction, I still think it is meaningful to explore more ways to replace the standard STE method.
5. The experiments are conducted not only on computer vision tasks but also on NLP and audio source separation.
6. Most previous quantization works are targeting a moderate accuracy range (mostly less than 80%) while models with very high accuracy can be harder to quantize. It's good to see that EfficientNetB3 and DeiT are also explored in the paper.

Weaknesses:
1. The DIFFQ method achieves marginal improvement over LSQ on ImageNet with different ResNets.
2. The analysis of why STE is sub-optimal is from other papers. Given that previous works have already tried to leverage the pseudo quantization noise, the novelty of this manuscript is not particularly strong.

---

### Review · Reviewer_eAuD · 2022-08-10

**Summary Of Contributions:**

This work presents a model compression method (DiffQ) based on additive noise injection during training, as an alternative to the widely used parameters quantization. Such Pseudo Quantization Noise (PQN) mimics the level of noise associated with quantization, while providing a differentiable pathway (with respect to both inputs and number of bits) for gradient propagation. PQN differs from common quantization techniques which rely on the non-differentiable *round()* operator to discretize the parameters and thus, during training backprop, are typically forced to bypass the operator by means of a Straight Through Estimator.

The paper shows that DiffQ outperforms other quantization strategies on 3 different tasks (image classification, language modeling, music source separation), achieving the most favorable trade-offs in terms of accuracy versus model compression.



**Broader Impact Concerns:**

Broader Impact Concerns are not discussed in the manuscript. I don't have specific concerns or requests on my side either.

**Requested Changes:**

As discussed in the previous section, my concerns at point 3 on supposed STE instabilities should be addressed, via a re-writing of the authors claims and/or with more conclusive experimental demonstrations and/or with better support from the literature.

I have a favorable view of this paper for all the reasons outlined in the previous section but, as it stands, these claims are not properly supported and must be fixed for me to recommend acceptance.

**Other minor changes**
- Some more comments on the effectiveness of gaussian noise would be valuable

- Quantization Aware Training refers to a class of techniques, while LSQ is a type of quantizer, which can be employed as part of QAT methods. In the various tables, what quantizers are used when results for QAT are reported? Are they symmetric / asymmetric, minmax quantizers, ...? It should be specified, at least as a table note / footnote.

- in fig.3 the label "bits bits" is quite confusing and a better name is warranted. I believe this is in reference to the total bits extracted from eq.12?

- I would recommend the authors to proofread the manuscript more carefully, as typos can be found, such as:
    - pag 7: compering
    - pag 10: symetric
    - pag 10: archiecture
    - pag 10: expanse

**Strengths And Weaknesses:**

**Relevance:** the topic of aggressive model compression at 8 bits and below discussed in this paper is of clear interest for the scientific community and industry alike.

**General comments:** the paper is well structured and easy to follow. The discussion of the literature properly cites and compares with closely related work. The implementation of the DiffQ method via PQN is adequately described and could be reproduced by a practitioner. The provided code is clean and well organized.

**Novelty:** noise injection at train time as a replacement for quantization has been previously reported (as acknowledged by the authors: see Baskin2018a,b), although only during part of the training. This work extends this concept by replacing quantization with PQN across the whole training and by using additive gaussian noise (instead of uniform or other forms). In addition, the authors leverage the differentiability w.r.t both inputs and bits to (1) allow for mixed precision (i.e., enable a flexible choice of number of bits for different layers / groups of parameters) and (2) explore a range of trade-offs between model size and accuracy. In my view, this clears the bar for novelty.

**Results:** solid results on image classification on a range of relevant models and datasets. Results on language models and music source separation are good but more limited. At least they add some degree of confidence on the applicability of the technique across different tasks.

**Weaknesses and concerns:**

1. DiffQ is only applied to weight quantization. This is effective for model compression but limits the applicability of this method to workload acceleration, which, in practical terms, requires the activations to be set to low-precision as well. However, it appears that recent work that builds up on the preprint of this very manuscript has already extended the work to activations as well (already cited: Park 2022), so my concerns are partially mitigated.

2. Due to hardware requirements, mixed precision strategies have limited applicability compared to uniform precision methods. DiffQ can also be applied with fixed quantization but results presented in the supplementary material appears not to show consistent improvement (see next point). This takes from the value of the proposed technique.

3. My main concern regards the claim that improved performance are to be attributed to having access to an end-to-end differentiable method, which avoids "STE instabilities". Although this claim is repeated throughout the manuscript, I don't think the authors have provided adequate evidence to support it. For example, fig 1b shows improved performance of DiffQ compared to 4 bit QAT but DiffQ relies on mixed precision while this instance of QAT does not, so the improvement can't be conclusively attributed to better handling of the backpropagation and better representation of the optimal weights by DiffQ. A fair comparison would be between models at fixed precision. When the authors did these experiments (Table B.4 suppl. material), results are not conclusive, *with DiffQ performing sometimes better, sometimes comparable, sometimes worse than their counterpart with regular quantizers set at an equivalent precision*. I can see the argument set in Section 3.3 (on a toy model), that at any given time QAT weights may be sub-optimal due to oscillations. I just don't think the improvements reported in this paper can be conclusively linked to this potential instability and not in part or entirely to the use of mixed precision. It should also be noted that, in my view, the papers cited in support of such "STE instabilities" tend to hand-wave the subject entirely or, alternatively, they address very aggressive quantization (2, 3 bits), which do not apply to the examples DiffQ is compared to in this manuscript (4,5,8 bits).

4. Use of gaussian noise instead of uniform noise (one of the main elements of novelty) appears to be really effective in improving performance but the authors provide very limited explanation of why this may be the case.

---

### Review · Reviewer_SyNn · 2022-08-17

**Summary Of Contributions:**

This paper proposed a noise injection method for differentiable quantization-aware training without STE. The proposed method, DiffQ, replaced the quantization operation on weights with a simple quantization noise injection modeled as a uniform random noise. The benefit of this simple derivation is that the model is differentiable with respect to weight as well as the bit-precision. Therefore, the authors further proposed a mixed-precision quantization mechanism where weights are grouped and quantized with different bit-precision. The authors provided extensive performance comparison with prior quantization methods.

**Broader Impact Concerns:**

I don't have specific concerns.

**Requested Changes:**

Please provide a more extensive ablation study and in-depth analysis of the three key techniques included in the proposed method to clarify their contributions to the improvement in the quantization performance.

**Strengths And Weaknesses:**

Strengths:
- Insightful example that differentiates DiffQ from QAT with STE
- Formulation of model-size aware optimization with grouping and mixed-precision
- Evaluation of various DNN applications including CNN and Transformer-based image classification, Transformer-based language model, and music source separation.

Weakness:
- There are several unsolved questions about the proposed techniques.
1) There are three claims in the proposed method: 1) noise injection for QAT without STE, 2) mixed-precision quantization derived by differentiable bit-width parameter, and 3) grouping of weights for fine-grain (mixed-precision) quantization. It is not clear how much each of these sub-techniques helps improve the model accuracy.
2) Table B.4 (Appendix) shows the accuracy comparison of the fixed-precision quantization. It is strange to see that DiffQ's accuracy is quite lower than LSQ for some cases (especially, ResNet18). Why does this happen?
3) Figure B.1(a) (Appendix) shows the accuracy comparison for different group sizes, but only for one network with very limited group size options. Noting that changing bit-precision in small granularity (g=1, 4, 8) is not practical, the author should consider experimenting with larger group size options, such as channel-wise grouping. Also, it is strange that the accuracy difference between g=inf and g=1 is marginal in Figure B.1(a). Is this a commonly observed phenomenon? If so, would group size matter?
4) Table B.7 (Appendix) shows that the quantization performance of different noise distributions. It is strange to see that the uniform distribution achieves much lower accuracy compared to the Gaussian since it contradicts the authors' assumption of quantization error modeling motivated by Widrwo et al., 1996. Does this experimental result imply that there is something wrong with the assumption of the error modeling?

---

### Decision · Action_Editors · 2022-09-20

**Recommendation:** Accept with minor revision

**Comment:**

The response from the authors and new revision of the paper addressed the bulk of the reviewers' concerns about this paper. After reading the most recent revision, I have two recommended revisions that will help make the paper clearer.
1. Make it clearer that variable bit-width quantization of models has value beyond inference time computational savings. From reading some of the authors' comments, I think that one goal of this work is to train models that are smaller to communicate to edge devices, even if the models are subsequently unpacked to larger bit widths. The reviewers seem to have assumed that the only value to reduced bit widths (or, at least, the primary value) is lower computational costs during inference.
2. Say a bit more about the added Gaussian noise. Specifically, the current revision does not say how the variance of the noise is selected. Is it set to match the variance of the more theoretically justified uniform noise in Eq. 6? Also, the **main paper should make it clear that the noise in all experiments is Gaussian.** Do not bury this detail in the appendix.

---

> ### Author Response · Authors · 2022-10-07
> **camera ready provided**
>
> Thanks for the acceptance and feedback. We have updated the camera ready accordingly. We added a sentence to say we use gaussian noise in practice in two places: when we introduce it, and at the top of the Results section. We added one sentence in the first paragraph of the intro to highlight that we aim at providing small models rather than seed up computation.